

# Ventilation of the Bay of Bengal oxygen minimum zone by the Southwest Monsoon Current

Peter M.F. Sheehan[1], Benjamin G.M. Webber[1,2], Alejandra Sanchez-Franks[3], and Bastien Y. Queste[4]

[1]Centre for Ocean and Atmospheric Sciences, School of Environmental Sciences, University of East Anglia, Norwich, United Kingdom
[2]Climatic Research Unit, School of Environmental Sciences, University of East Anglia, Norwich, United Kingdom
[3]National Oceanography Centre, Southampton, United Kingdom
[4]Department of Marine Sciences, University of Gothenburg, Gothenburg, Sweden

**Correspondence:** Peter Sheehan (p.sheehan@uea.ac.uk)

**Abstract.** Oxygen minimum zones occupy large areas of the tropical subsurface oceans and substantially alter regional bio-geochemical cycles. In particular, the removal rate of bio-available nitrogen (de-nitrification) from the water column in oxygen minimum zones is disproportionate to their size. The Bay of Bengal is one of the strongest OMZs in the global oceans; however, variable sources of oxygen prevent the onset of large-scale de-nitrification. The various oxygen-supply mechanisms that

maintain oxygen concentrations in the OMZ above the denitrification threshold are currently unknown. Here, using a combination of multi-platform observations and model simulations, we identify an annual supply of oxygen to the Bay of Bengal in the high-salinity core of the Southwest Monsoon Current, a seasonal circulation feature that flows northwards into the Bay during the South Asian southwest monsoon (i.e. June to September). Oxygen concentrations within the Southwest Monsoon Current (80 to 100 $\mu$mol kg$^{-1}$) are higher than those of waters native to the Bay (i.e. < 20 $\mu$mol kg$^{-1}$). These high-oxygen waters

spread throughout the central and western Bay of Bengal, leading to substantial spatio-temporal variability in observed oxygen concentrations. Moreover, the northward oxygen transport of the Southwest Monsoon Current is a spatially and temporally distinct event that stands out from background oxygen transport. Models indicate that variability in annually integrated oxygen supply to the BoB varies with the strength of the Southwest Monsoon Current more closely than with its oxygen concentration. Consequently, we suggest that predictability of the annual oxygen flux is likely aided by understanding and predicting the

physical forcing of the Southwest Monsoon Current. Our results demonstrate that the current, and in particular its high-salinity, high-oxygen core, is a feature relevant to the processes and communities that drive denitrification within the Bay of Bengal that has heretofore not been considered.

## 1 Introduction

Oxygen minimum zones (OMZ) are intermediate-depth (i.e. around 300 to 1000 m) regions of the ocean that contain very

little dissolved oxygen – below 60 $\mu$mol kg$^{-1}$ (Hofmann et al., 2011) – and in which macrofauna struggle to survive. The atypical chemical pathways of these regions play an important role in biogeochemical cycling, exerting an influence on the global cycles of nitrogen and carbon (Bange et al., 2005; Gruber, 2008) that is disproportionate to their size (Johnson et al.,





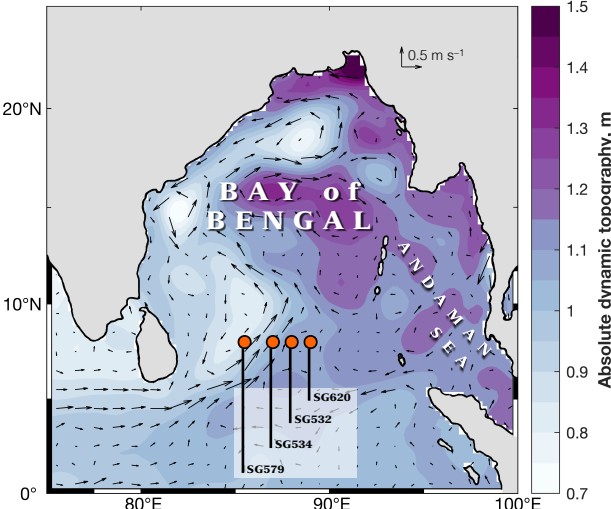

**Figure 1.** Mean absolute dynamic topography (m; shading) and surface geostrophic velocity (m s$^{-1}$; vectors) from satellite altimetry, averaged over July 2016. The locations of the Seagliders used in this study are indicated by the orange circles.

2019). The solubility of oxygen in sea water decreases with temperature and, globally, the oxygen content of the oceans has declined over the last six decades (Kwiatkowski et al., 2020); however, predicting the fate of OMZs in a warming 21st-century

ocean is not straightforward. While OMZs have generally expanded since the middle of the last century (e.g. Zhou et al., 2022), recent work has pointed to a varied future response to climate change across different regions of OMZs: the outermost layer, which is highest in oxygen, increases in volume even as the innermost, low-oxygen core decreases in volume (Busecke et al., 2022; Ditkovsky et al., 2023).

OMZs occur throughout the tropical ocean; in the Indian Ocean, prominent OMZs are found in the Arabian Sea and the

Bay of Bengal (Fig. 1), semi-enclosed basins open only along their southern boundaries. Of these, the Bay of Bengal, in which oxygen concentrations are generally below 10 $\mu$mol kg$^{-1}$ at depths between 100 and 500 m (D'Asaro et al., 2020), is thought to be intensifying and approaching a tipping point, beyond which the rate of denitrification might accelerate (Bristow et al., 2017). The Bay is characterised by strong stratification, a consequence of high freshwater input from river runoff and monsoon precipitation. Because this haline stratification limits vertical oxygen fluxes (Rixen et al., 2020), lateral advection is

a key component of the Bay's oxygen budget (Johnson et al., 2019). Bio-Argo observations have strongly suggested a role for physical processes at the Bay's open southern boundary: the sporadic injection of relatively oxygen-rich water is thought to provide sufficient oxygen to hinder denitrification and to generate a system that is highly variable in time and space (Johnson et al., 2019; D'Asaro et al., 2020; Sheehan et al., 2020). Previous work has emphasised the potential role of eddies is driving such an oxygen injection (Johnson et al., 2019; D'Asaro et al., 2020), although their impact is difficult to quantify, not least

because they can contribute to both the supply and consumption of oxygen (Rixen et al., 2020). The role of the Bay's basin-scale circulation, and in particular the influence of its pronounced seasonal variability, on the OMZ has received less attention,





despite the documented influence of basin-scale circulation in defining the intensity and outline of the neighbouring Arabian Sea OMZ (Lachkar et al., 2023; Font et al., 2024).

The most prominent basin-scale features of the Bay of Bengal are its seasonally reversing monsoon currents. Between
January and March, the Northeast Monsoon Current flows southwards off the eastern coast of Sri Lanka; between June and September, the situation is reversed: the Southwest Monsoon Current (SMC) flows northward in the same region (Fig. 1; Vinayachandran et al., 1999; Webber et al., 2018). This latter current, forced initially by local wind stress at the ocean surface, entrains recently ventilated Arabian Sea High-Salinity Water (ASHSW) from the Indian Ocean's equatorial current systems (Sanchez-Franks et al., 2019). The horizontal pressure gradient between the ASHSW's core, which is denser than the low-
salinity waters native to the Bay of Bengal, further strengthens the geostrophic flow of the SMC. Consequently, the high-salinity core of ASHSW is found on the eastern flank of the northward-flowing SMC, which reaches a peak where the horizontal salinity gradient is strongest (Webber et al., 2018).

The SMC has been found to advect trace amounts of oxygenated Persian Gulf Water into the Bay of Bengal's OMZ (Sheehan et al., 2020). But besides the relatively minor oxygen contribution of Persian Gulf Water: (1) the oxygen transport of the
relatively oxygen-rich ASHSW and (2) the extent to which ASHSW penetrates into and circulates around the Bay, have not been assessed. Here, we present ocean glider observations from across the SMC that demonstrate the high oxygen content of the ASHSW – especially in comparison to the low oxygen content of the OMZ. Further, we analyse particle tracking experiments conducted using an ocean re-analysis product to determine the spread of this high-salinity, high-oxygen water within the Bay of Bengal, as well as demonstrating the geographic origin of the Bay's high- and low-salinity waters. Identifying transport and
pathways of oxygenated inflows can help shed light on the the processes limiting denitrification in the Bay of Bengal OMZ.

## 2   Data and methods

### 2.1   Observations

Hydrographic observations were collected using four Seagliders (Eriksen et al., 2001) in the southwestern Bay of Bengal as part of the Bay of Bengal Boundary Layer Experiment (Vinayachandran et al., 2018; Webber et al., 2019). The gliders were
deployed east to west along 8°N, at 86, 87, 88 and 89°E. Deployed at 86°E, SG579 transited to 85.3°E after seven days (38 dives), where it stayed for the remaining 12 days of the deployment (78 dives). Otherwise, the gliders were operated as "virtual moorings": i.e. they remained on-station for the entire deployment. Observations were processed to optimise the hydrodynamic model of the glider's flight path (Frajka-Williams et al., 2011) and to correct for thermal lag of the un-pumped conductivity cell (Garau et al., 2011). Hydrographic observations were interpolated in time and potential density using Gaussian windows
of six hours and 0.25 kg m$^{-3}$ respectively.

We supplement our glider observations with:

1. **The World Ocean Atlas** observation-based climatology (Fig. 1a; Locarnini et al., 2024; Reagan et al., 2024; Garcia et al., 2024), which we interpolate onto isopycnals.





2. **Satellite altimetry** observations of surface velocity, provided by the Copernicus Marine Data Service (Product ID: SEALEVEL_GLO_PHY_L4_NRT_008_046).

3. **Bio-Argo float** observations. Salinity, oxygen and pressure corrections were applied to the Bio-Argo float observations following (Bittig et al., 2018). Oxygen observations were compared to World Ocean Atlas surface observations to correct for storage drift following (Bittig et al., 2019). We identify spikes in the Argo observations associated with high-salinity, high-oxygen water on the 24 kg m$^{-3}$ isopycnal using the criteria of absolute salinity $> 35$ g kg$^{-1}$ and oxygen concentration $> 30$ $\mu$mol kg$^{-1}$ corresponding to properties of the SMC. These criteria were selected to best match the spikes identified in the Argo float observations on this isopycnal.

## 2.2 NEMO-MEDUSA biogeochemical model

We extend and contextualise our observations using output from NEMO-MEDUSA, a coupled biogeochemical-physical model. The physical simulation uses version 3.6 of the global-ocean NEMO model (Nucleus for European Modelling of the Ocean; Madec, 2008) and is forced with re-analysed atmospheric data from the Drakkar Surface Forcing dataset version 5.2 (Brodeau et al., 2010). The physical model is coupled to the biogeochemical model MEDUSA-2 (Yool et al., 2013a). NEMO-MEDUSA has 1/12° resolution in latitude and longitude, and 75 vertical levels, the upper 35 of which (i.e. to 300 m) are within the depth range of the Southwest Monsoon Current; vertical resolution decreases with depth. The physical model is run from 1958 and is coupled to the biogeochemical model from 1990. Here, we use monthly mean output from 1998 to 2011 in order to exclude a period of spin-up of the biogeochemical model in the eight years after coupling. Physical and biogeochemical output are available at monthly resolution.

We calculate meridional oxygen transport, $T_{O_2}$ (mol s$^{-1}$), in NEMO-MEDUSA across 8°N between the 24 and 26 kg m$^{-3}$ (potential density) isopycnals according to:

$$T_{O_2} = [O_2] \cdot v \cdot \Delta x \cdot \Delta z \cdot \rho \cdot 10^{-6} \qquad (1)$$

where $[O_2]$ is oxygen concentration ($\mu$mol kg$^{-1}$) and $v$ (m s$^{-1}$) is meridional velocity, both averaged between the 24 and 26 kg m$^{-3}$ isopycnals, $\Delta x$ is the horizontal width of each model grid cell (m), $\Delta z$ is the vertical distance between the 24 and 26 kg m$^{-3}$ isopycnals (m), and $\rho$ is density (kg m$^{-3}$); the final two terms of Equation 1 convert oxygen concentration from $\mu$mol kg$^{-1}$ to mol m$^{-3}$. The 24-to-26 kg m$^{-3}$ density range covers the SMC and its high-salinity core; by using the 24 kg m$^{-3}$ isopycnal as the upper limit, we exclude the mixed layer and thus waters with an oxygen concentration that is in equilibrium with the atmosphere. Finally, we calculate the oxygen flux, $F_{O_2}$ (mol; across 8°N, across the width of the BoB and between the 24 and 26 kg m$^{-3}$ isopycnals), at annual resolution by integrating $T_{O_2}$ over both the cross-sectional area of the SMC (i.e. longitude and thickness) and the period of each year's southwest monsoon (i.e. June to September inclusive).





### 2.2.1 Model validation

The NEMO-MEDUSA coupled model employed here has been used extensively globally (e.g. Yool et al., 2013a, b; Popova
et al., 2016), including in the Indian Ocean (e.g. Jacobs et al., 2020; Jebri et al., 2020; Asdar et al., 2022). The SMC is
present as a northward flow in the southwestern Bay of Bengal, and the basin-scale distribution of oxygen is reasonably
well reproduced: oxygen concentrations are highest between the equator and approximately 5°N, and are lower within the
Bay proper (Fig. 2a). Immediately north of the equator, modelled oxygen concentrations of approximately 120 $\mu$mol kg$^{-1}$
compare favourably to the World Ocean Atlas observational climatology (Fig. 2). However, the model does not manage to
fully reproduce the intensity of the BoB OMZ, at least in an average July: north of approximately 10°, climatological observed
oxygen concentrations are everywhere in the region of 20 $\mu$mol kg$^{-1}$ (Fig. 1a) while modelled oxygen concentrations are
between 20 and 80 $\mu$mol kg$^{-1}$ higher than this (Fig. 2b and c). The larger differences are associated with roughly circular
regions of elevated oxygen concentration found in the western Bay of Bengal in the model. In addition, the glider observations,
in which we find ambient oxygen concentrations below 20 $\mu$mol kg$^{-1}$ (Fig. 3), similarly suggest that oxygen concentrations
in the Bay of Bengal as simulated by NEMO-MEDUSA are too high. OMZ concentrations in the model representation suffer
from inadequate representation of biogeochemical cycling and the representation of sub-mesoscale or ageostrophic processes
(Armstrong McKay et al., 2021; Lévy et al., 2024); however, our analysis focuses on the physical representation of the SMC,
rather than the biogeochemical properties in the model, similar to other such studies of the Indian Ocean (Queste et al., 2018).

### 2.3 Particle-tracking experiments

We conduct particle tracking experiments using the GLORYS12 ocean physics re-analysis (Lellouche et al., 2021, product
ID: GLOBAL_MULTIYEAR_PHY_001_030), covering the 26-year period between 1994 and 2019 (inclusive). This NEMO-
based re-analysis product has the same same horizontal and vertical resolution as the NEMO-MEDUSA simulation described
above, and is available at daily resolution. It has been shown to compare particularly well to physical observations in the
Indian Ocean (Webber et al., 2018; Sheehan et al., 2020) and has been used previously in particle tracking experiments in
the region by Sanchez-Franks et al. (2019); Sheehan et al. (2020), whose methods we repeat here. Salinity and velocity fields
from the re-analysis are interpolated onto the 24 kg m$^{-3}$ isopycnal – i.e. the core density of ASHSW previously used when
tracing this water mass (Sanchez-Franks et al., 2019) – on which the particle tracking experiments are then conducted. Re-
analysis velocities are bi-linearly interpolated onto the particle locations, and the particles are advected forwards or backwards
as desired using a fourth-order Runge-Kutta scheme with a time step of 24 hours. This method determines advective transport
along an isopycnal and does not account for isopycnal or diapycnal diffusivity.

We first conduct forwards-tracking experiments to determine the distribution of the SMC's high-salinity core within the Bay
of Bengal. Particles are initialised daily between 1 June and 30 September inclusive, those being the months during which the
SMC is present (Webber et al., 2018), and only where salinity is greater than 35.2 PSU. (Note that the GLORYS12 re-analysis
outputs practical salinity in PSU.) Given this salinity constraint, the number of particle released each day is not constant.



Particles are initialised every 0.1° longitude across 8°N, between the eastern coast of Sri Lanka and 88°E. Particles that travel south of 6°N are removed.

Secondly, and to determine the origin of the oxygen-rich waters, we conduct backwards-tracking experiments in which particles are initialised at five-day intervals, backwards from 31 December 2020 to 1 January 1994, and on a 0.2° grid between 82 and 92°E, and 10 and 15°N. In separate experiments, particles are initialised at grid points with: (1) salinity greater than

35.2 PSU; and (2) salinity less than 35 PSU. Individual particles are tracked backwards for up to one year.

## 3 Results

### 3.1 Observations of the Southwest Monsoon Current

Between 85.3 and 86°E (SG579), on the western edge of the SMC, there is no evidence of the high-salinity core (Fig. 3), in line with previous observations and the dynamics of the SMC (Webber et al., 2018). From 22 to 26 kg m$^{-3}$, salinity increases

from 34.7 to 35.1 g kg$^{-1}$. The situation is markedly different further east: all three remaining gliders capture the high-salinity core of the SMC. The gradual and steady decrease in salinity with depth in ambient waters – i.e. as observed by SG579 – is interrupted by a pronounced increase in salinity centred on the 24 kg m$^{-3}$ isopycnal (Fig. 3b–d). Peak salinity values within the high-salinity core are in excess of 35.6 g kg$^{-1}$. Below the core, salinity decreases rapidly: at the 26 kg m$^{-3}$ isopycnal, the salinity values are similar to those observed in the west where the core is absent (Fig. 3).

The distribution of oxygen is also markedly different in the presence of the high-salinity core. In the west, outside of the core, oxygen decreases with depth from 100 to below 20 $\mu$mol kg$^{-1}$ (Fig. 3a). Within the core, oxygen levels are elevated and increase with increasing salinity: the highest salinities are associated with oxygen concentrations between 60 and 100 $\mu$mol kg$^{-1}$ (Fig. 3b–d). We note that these oxygen concentrations are an order of magnitude greater than background concentrations within the BoB OMZ (D'Asaro et al., 2020). At the base of the high-salinity core, contours of oxygen follow contours of salinity par-

ticularly closely (Fig. 3b–d). Further, the temporal discontinuity in the high-salinity core observed at 87°E (SG534) makes the close relationship between salinity and oxygen within the core particularly clear: when the core is absent, the distributions of oxygen and salinity resemble those observed in the west between 85.3 and 86°E (Fig. 3a and b).

Apparent oxygen utilisation (AOU; i.e. the difference between concentration at saturation and the observed oxygen concentration) further demonstrates the difference between background water masses and the high-salinity core (Fig. 3). As oxygen

is less soluble in the warm, salty waters of the high-salinity core, the difference in oxygen concentration is less apparent than with AOU which reveals the actual deficit in oxygen in these water masses. This AOU gradient is evidence of the general lack of ventilation and high respiration that occurs in the sub-surface waters of the OMZ; the substantially lower AOU in the high-salinity core suggests more recent subduction of this water mass.



## 3.2 Oxygen transport of the Southwest Monsoon Current

We use the NEMO-MEDUSA physical-biogeochemical model to investigate the seasonal and longitudinal variability of oxygen transport into the BoB. Meridional oxygen transport at 8°N in the Bay of Bengal is generally low (between $-2$ and $2 \times 10^3$ mol s$^{-1}$), save for: (1) a narrow, near year-round strip of pronounced southward transport in the far west (82°E); and (2) a region of pronounced northward transport to the immediate east of this during the four months of the southwest monsoon (83 to 87°; Fig. 4a). This latter feature is clearly associated with the SMC: both its longitudinal and temporal extent match the

northwards velocities associated with the SMC, being enclosed within the 0.1 m s$^{-1}$ velocity contour (Fig. 4a). Furthermore, the gradual westward movement of the SMC's core over the duration of the southwest monsoon matches observations (Webber et al., 2018).

Northward oxygen transport increases markedly between May and June as the SMC strengthens (Fig. 4a). Transport peaks at $12 \times 10^3$ mol s$^{-1}$ in July at about 85°E; it decreases to around $6 \times 10^3$ mol s$^{-1}$ in August – although this is still considerably

higher than background values – before attaining a second peak of $12 \times 10^3$ mol s$^{-1}$ in September at about 83°E (Fig. 4a). The July peak in oxygen transport is associated with a peak in the SMC's meridional velocity (i.e. $> 0.2$ m s$^{-1}$; Fig. 4a). The September peak in oxygen transport is associated with lower meridional velocity than the July peak, but with a higher oxygen concentration (Fig. 4b). Oxygen concentration in the SMC at 8°N is elevated relative to background levels, but there is a month's lag between the increase in velocity and the increase in oxygen concentration (Fig. 4b). Furthermore, maximum oxygen

concentrations are located approximately one degree east of the velocity maximum. This mirrors the salinity distribution: the SMC's high-salinity core is one degree or so east of the velocity core (Webber et al., 2018; Sanchez-Franks et al., 2019) and, as noted above, it is the salinity gradient between the core and low-salinity ambient waters further west that in part drives the northward flow. This agreement in the location of the oxygen and salinity maxima is therefore consistent with the correlation between salinity and oxygen concentration in the inflowing water identified in the glider observations (Fig. 3).

The prominence of the SMC as an advective source of oxygen to the BoB OMZ is clear in both space and time. The SMC drives a seasonal increase in the longitudinal-mean oxygen transport, which peaks at $1.57 \times 10^3$ mol s$^{-1}$ in July (Fig. 4c). Outside of the southwest monsoon, longitudinal-mean oxygen transport is near zero or even negative (i.e. southward). Furthermore, the time-mean oxygen transport peaks across the longitudes of the SMC (Fig. 4d).

Oxygen is not a conservative tracer, and the oxygen content of the BoB OMZ depends on the interplay of biogeochemical

and physical processes; but the annual cycle of oxygen content of the BoB OMZ is consistent with an injection of oxygen by the SMC. Oxygen content between the 24 and 26 kg m$^{-3}$ isopycnals is lower before the southwest monsoon than after, both according to NEMO-MEDUSA and the World Ocean Atlas. In the observations, oxygen content increases from a near minimum in July ($8.25 \times 10^{10}$ mol) to a maximum in September ($11.19 \times 10^{10}$ mol; Fig. 5). The July-to-September increase is 31% of the annual mean. The annual cycle in NEMO-MEDUSA is smoother and has a reduced amplitude compared to

observations, but it similarly indicates an increase in oxygen content between the 24 and 26 kg m$^{-3}$ isopycnals during the southwest monsoon (Fig. 5; standard deviations are not plotted for World Ocean Atlas observations, which are available only as climatological monthly means).



The annual oxygen flux during the southwest monsoon, between the 24 and 26 kg m$^{-3}$ isopycnals across 8°N, is commonly around 1 to 2 × 10$^{12}$ mol – save for the exceptionally large flux in 1998 (3.1 × 10$^{12}$ mol; Fig. 6a; we note that there was
a particularly strong El Niño event and pronounced anomalies in the Indian Ocean in 1997 and 1998; Murtugudde et al., 2000). In an average year, between these isopycnals, the SMC represents the dominant oxygen flux event at 8°N, even across the full width of the Bay (Fig. 4a). The magnitude of the oxygen flux into the Bay of Bengal varies more strongly with the volume transport across 8°N (averaged between 24 and 26 kg m$^{-3}$ over the southwest monsoon; $R^2$ = 0.56; Fig. 6b) than with the oxygen concentration of the in-flowing water (also averaged between 24 and 26 kg m$^{-3}$ over the southwest monsoon;
$R^2$ = 0.24; Fig. 6c). This would appear to be consistent with the results presented above: that the location of the strongest oxygen flux within the SMC matches the location of the strongest northward velocity, with the highest oxygen concentrations being located, like the highest salinities, to the east of this velocity maximum.

### 3.3 Spreading of Southwest Monsoon Current water in the Bay of Bengal

We conduct forwards particle tracking experiments performed using the GLORYS12 ocean reanalysis data to determine the
distribution of the high-salinity, high-oxygen water of the SMC as it spreads across the Bay of Bengal. In the forward-tracking particle experiment, where particles are initialised at grid points with a salinity greater than 35.2 PSU, the presence of SMC water in the Bay increases steadily throughout the southwest monsoon (i.e. June to September; Fig. 7). In June, the particles cover a small area in the immediate vicinity of the SMC, before spreading out to cover the entire southwestern portion of the Bay by September (Fig. 7a–d). The density of particles begins to decrease from November (Fig. 7e), even as particles penetrate
further northward. By May, immediately before the onset of the following southwest monsoon, particles are clustered in a broad region in the west-central BoB, off the eastern coast of India (Fig. 7l). At no point in the year does SMC-origin water significantly penetrate the Andaman Sea – i.e. the eastern portion of the BoB basin (Fig. 7). The overall distribution of particles demonstrates the northward spread of water in the SMC's high-salinity core within the Bay; it reaches all but the most northern and eastern parts of the Bay.

There is broad agreement between the regions of the Bay of Bengal in which SMC-origin water is found and the locations in which relatively high-oxygen, high-salinity spikes (i.e. salinity > 35 g kg$^{-3}$ and oxygen > 30 $\mu$mol kg$^{-1}$; spikes hereafter) are found on the 24 kg m$^{-3}$ isopycnal in bio-Argo profiles (Fig. 7). (Note that the thresholds for identifying spikes in the bio-Argo float data within the BoB are lower than the thresholds used in the SMC region; these lower thresholds allowed for spikes to be identified further north in the BoB where the signal is eroded due to mixing with the surrounding water masses.)
Spikes are confined to the very south in June and July, and not in any Bio-Argo profiles in the central and northern Bay. The area in which high-salinity, high-oxygen spikes are found expands as SMC-origin water spreads northward between August and November (Fig. 7c–f); and the percentage of bio-Argo profiles with spikes increases markedly during months in which the amount of SMC-origin water is at its highest (Fig. 7). The high-oxygen, high-salinity signal of SMC-origin water must therefore be preserved up to this time as the water spreads in the Bay. Between December and May, most bio-Argo profiles
do not contain spikes (Fig. 7g–l), even though many profiles are from regions in which SMC-origin water may be found. This





suggests that the high-oxygen, high-salinity signature of the SMC within the Bay of Bengal becomes eroded from December onwards, in agreement with the seasonal cycle of oxygen content within the OMZ (Fig. 5).

We conduct backwards particle tracking experiments to determine source regions of the relatively low-salinity ($< 35$ PSU) and relatively high-salinity ($> 35.2$ PSU) waters found on the 24 kg m$^{-3}$ isopycnal in the central Bay of Bengal. Low-salinity particles originate and circulate predominantly within the Bay, with high track density even reaching into the northern Andaman Sea (Fig. 8a). This is in contrast to the high-salinity particles, which have a higher track density in the south-western Bay than in the northern Bay (Fig. 8b). Furthermore, more high-salinity particles originate in the western equatorial Indian Ocean, off the coast of East Africa, than do low-salinity particles; this is consistent with trajectories of high-salinity SMC water previously identified (Sanchez-Franks et al., 2019). Very few particles, of either high or low salinity, originate from the southeastern Indian Ocean (Fig. 8). These results demonstrate the separation in the Bay of Bengal's source waters: the high-salinity, high-oxygen water is transported from the southwest and much of it originates in the southeastern Arabian Sea or western Indian Ocean; and the SMC is likely a key conduit for this water. Meanwhile, the relatively fresh water found within the central Bay of Bengal is more likely native to the Bay, particularly given that the large influx of river water that occurs after each years' southwest monsoon.

## 4  Discussion

The high salinity core of the SMC has been the subject of previous investigation (e.g. Vinayachandran et al., 2013; Sanchez-Franks et al., 2019), but its high oxygen transport has not previously been identified. Indeed, it has been suggested that ASHSW entering the Bay of Bengal is low in oxygen and so is unlikely to contribute to the oxygen budget of the OMZ (Rixen et al., 2020). Our observations suggest the contrary: the high-salinity core of the SMC is markedly less depleted in oxygen than the waters native to the Bay of Bengal. This high-oxygen core represents a considerable advective oceanic transport of oxygen to the upper Bay of Bengal OMZ, with consquences on the OMZ's annual cycle of oxygen. Ditkovsky et al. (2023) have recently highlighted the importance of outflows from marginal seas, similar to the SMC in size and transport, for defining the expansion-redistribution-contraction behaviour observed in OMZs. Our work provides clear observational evidence of how mesoscale advective currents, not similar to marginal sea outflows, can play an important role in ventilating the upper boundary of the BoB OMZ.

We suggest that both the intensity and location of the SMC's oxygen signature are determined by physical processes more strongly than by biogeochemical processes: the peak in meridional oxygen transport is co-located not with oxygen concentration but with meridional velocity (Fig. 1); and the magnitude of the SMC's oxygen flux in a given year depends more on that current's volume transport than on the oxygen concentration of its ASHSW core (Fig. 6). Consequently, it appears that processes which drive variability in the oxygen concentration ASHSW have only a modulating effect on the SMC's oxygen flux, and that the predictability of the SMC's oxygen flux derives primarily from understanding and predicting the SMC's physics. Hence, earth system models used to predict the future state of the Bay of Bengal's OMZ under climate change must correctly represent the physical forcing mechanisms of the local and dynamic SMC – for instance, local wind forcing, and



Kelvin and Rossby waves (Webber et al., 2018). Nevertheless, significant future changes to the oxygen concentration of the
SMC's ASHSW core will necessarily influence oxygen delivery to the Bay of Bengal, although the evolution of the oxygen
concentration of ASHSW remains highly uncertain: opposing trends are predicted over different timescales and significant
disagreement between models (Lachkar et al., 2023). Oxygen concentration changes could result from changes to the upstream
respiration rate within ASHSW. While the waters of the SMC core are more oxygenated than those native to the Bay of Bengal,
the ASHSW core is nevertheless under-saturated (Fig. 3) and has therefore been subducted for some time. Prolonged residence
time, perhaps as a result of circulation changes, or increased particulate export and respiration upstream before ASHSW enters
the Bay of Bengal would reduce the oxygen content of the SMC.

The identification of the considerable oxygen delivery to the Bay of Bengal by the SMC places this seasonal inflow as a can-
didate phenomenon for maintaining the oxygen concentration of the OMZ above the level at which large-scale de-nitrification
occurs – and below which biogeochemical pathways within the OMZ would be significantly altered. The onset of large-scale
de-nitrification within the Bay of Bengal would likely have global ramifications, and is thought to be presently avoided by even
fairly modest injections of oxygenated water (Bristow et al., 2017; Johnson et al., 2019). Constructing an oxygen budget for
the Bay of Bengal is beyond the scope of the present contribution – clearly this will depend on many more processes, both
physical and biogeochemical, than on just the one considered here. But while previous work has emphasised the importance
of eddies as an means by which oxygen is supplied to the Bay of Bengal OMZ (Queste et al., 2018; Johnson et al., 2019), our
results suggest that the SMC, and in particular its core of ASHSW, should be considered an additional means by which oxygen
is supplied to the OMZ.

## 5   Conclusions

The Southwest Monsoon Current (SMC) flows northward into the Bay of Bengal during the four months of the South Asian
southwest monsoon (i.e. June to September). The SMC advects into the Bay a core of Arabian Sea High-Salinity Water
(ASHSW), the presence of which further, through geostrophy, intensifies the SMC to the core's immediate west. The present
observations have demonstrated that this core of ASHSW is also high in oxygen: it's characteristic oxygen concentration
of 80 to 100 $\mu$mol kg$^{-1}$ is much higher than that of those waters found at similar densities that are native to the Bay (i.e.
$< 20$ $\mu$mol kg$^{-1}$). Moreover, the northward oxygen transport of the SMC is an annual event that stands out as such in both
temporal and spatial averages: at no other time of year and at no other longitude, from India to Thailand, is there an advective
oxygen event of comparable duration or magnitude. The high-salinity, high-oxygen waters of the SMC spread throughout the
Bay of Bengal, and their origin is largely different to that of low-salinity waters also found in the Bay. Given the proximity of
the Bay of Bengal's OMZ to the de-nitrification threshold, the SMC and its attendant oxygen supply may well be a key feature
of biogeochemical cycles in the world ocean.

Year to year, the magnitude of the SMC's annually integrated oxygen flux varies with physical factors (i.e. volume transport)
much more closely than with the oxygen concentration of the inflowing ASHSW. Consequently, present and future predictabil-
ity of the SMC's annual oxygen flux likely lies in understanding and predicting the physical forcing of the current itself.





Nevertheless, attributing the high-oxygen signal of the inflowing water to ASHSW highlights the importance of understanding the oxygen dynamics within both the Arabian Sea source region and equatorial transport pathways over which biogeochemical changes might influence oxygen concentration in the future.

*Data availability.* Seaglider observations (Webber et al., 2019) are available from the British Oceanographic Data Centre (*www.bodc.ac.uk*) under DOI: *doi:10.5285/996bf53d-5448-297a-e053-6c86abc0b996*. The code for MEDUSA-2 (Yool et al., 2013a) is available from the repository of Geoscientific Model Development: *gmd.copernicus.org/articles/6/1767/2013/gmd-6-1767-2013-supplement.zip*. The GLO-RYS12 ocean physics re-analysis (Lellouche et al., 2021) is available from the Copernicus Marine Data Centre (*marine.copernicus.eu*); we here use GLOBAL_MULTIYEAR_PHY_001_030 which is available under the DOI: *doi.org/10.48670/moi-00021*. Satellite altimeter-
derived surface currents are also available from the Copernicus Marine Data Centre; we here use SEALEVEL_GLO_PHY_L4_MY_008_047 which is available under the DOI: *doi.org/10.48670/moi-00148*. The observation-based climatology from the World Ocean Atlas 2023 (Locarnini et al., 2024; Reagan et al., 2024; Garcia et al., 2024) is available at: *www.ncei.noaa.gov/data/oceans/woa/WOA18/DATA*.

*Author contributions.* BGMW conceived the research. PMFS and BGMW conducted the research with input from ASF and BYQ. PMFS wrote the paper, with input and revisions from all authors.

*Competing interests.* The authors declare that they have no competing interests.

*Acknowledgements.* PMFS, BGMW and ASF were supported by the Bay of Bengal Boundary Layer Experiment (PMFS: NE/L013827/1; BGMW: NE/L013827/1; ASF: NE/L013835/1), a joint NERC/UKRI–Ministry of Earth Sciences (UK/India) project. PMFS was further supported by the ERC-funded COMPASS project (741120). ASF was also supported by the NERC/UKRI-funded NEW-NORMAL project (NE/W003813/1). BYQ was supported by ONR GLOBAL Grant N62909-21-1-2008 and the Swedish Formas Grant 2022-01536.



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





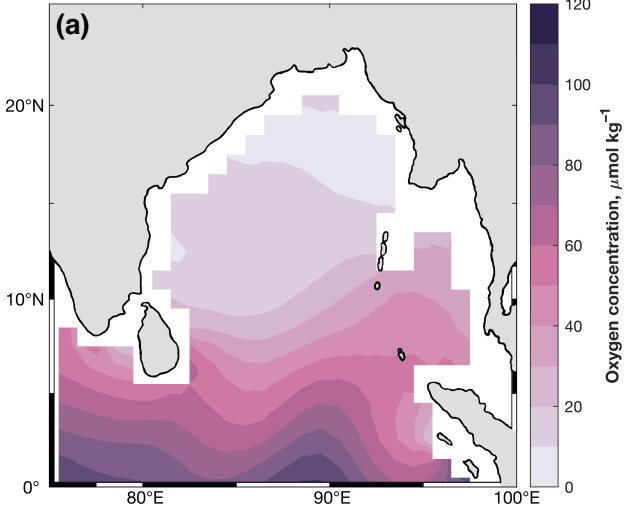

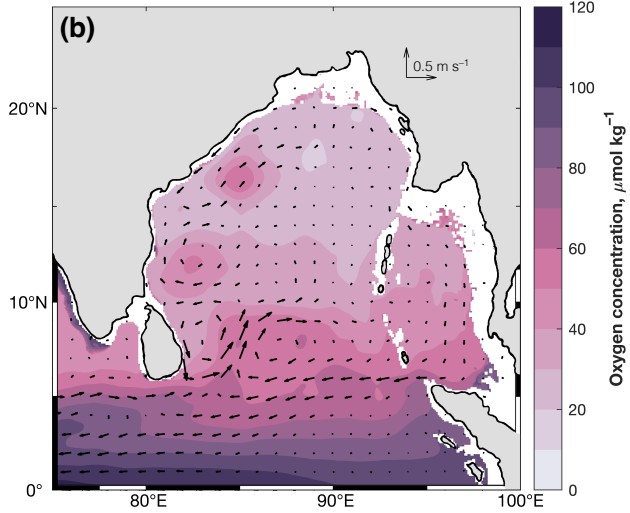

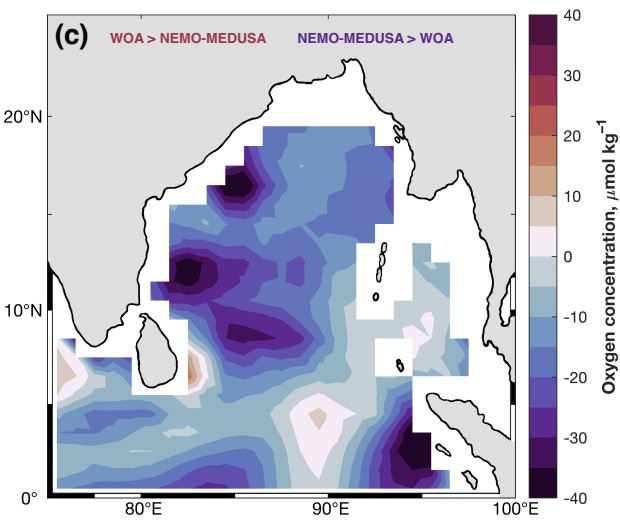





**Figure 2. (a)** Mean July oxygen concentration ($\mu$mol kg$^{-1}$; shading) between the 24 and 26 kg m$^{-3}$ isopycnals from the World Ocean Atlas. **(b)** Mean July oxygen concentration ($\mu$mol kg$^{-1}$; shading) and velocity (m s$^{-1}$; vectors), similarly between the 24 and 26 kg m$^{-3}$ isopycnal, from NEMO-MEDUSA. **(c)** The difference ($\mu$mol kg$^{-1}$) between mean July oxygen concentration in the World Ocean Atlas and NEMO-MEDUSA (former minus latter).



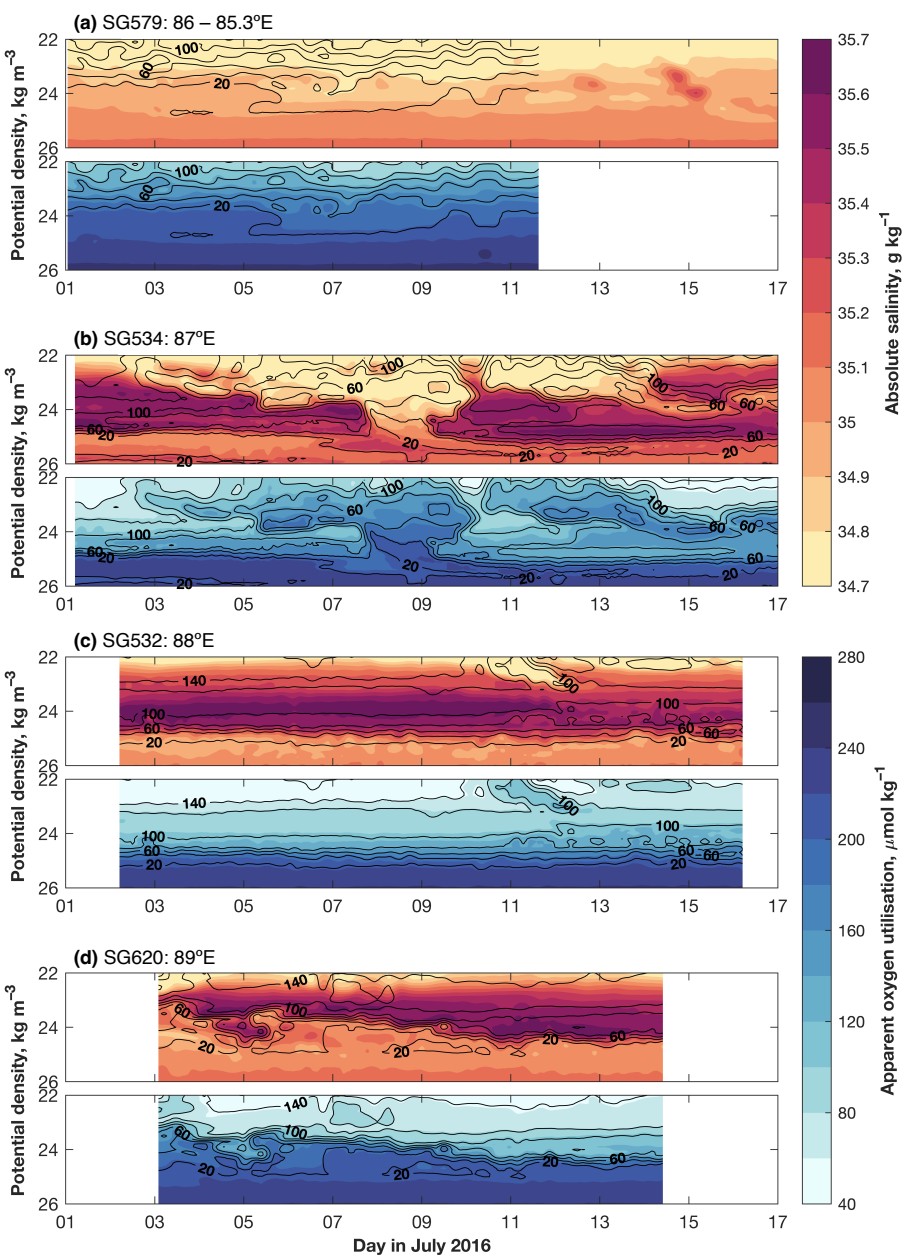

**Figure 3.** Time series of glider observations from **(a)** SG579, **(b)** SG534, **(c)** SG532 and **(d)** SG620. In each pair of panels, orange-coloured shading in the *upper panel* is absolute salinity (g kg$^{-1}$), and blue-coloured shading in the *lower panel* is apparent oxygen utilisation ($\mu$mol kg$^{-1}$); in all panels, contours are oxygen concentration ($\mu$mol kg$^{-1}$). Time series are plotted against potential density (kg m$^{-3}$). The location of each glider is shown in Fig. 1. Note that the oxygen sensor on SG579 malfunctioned on 11 July.

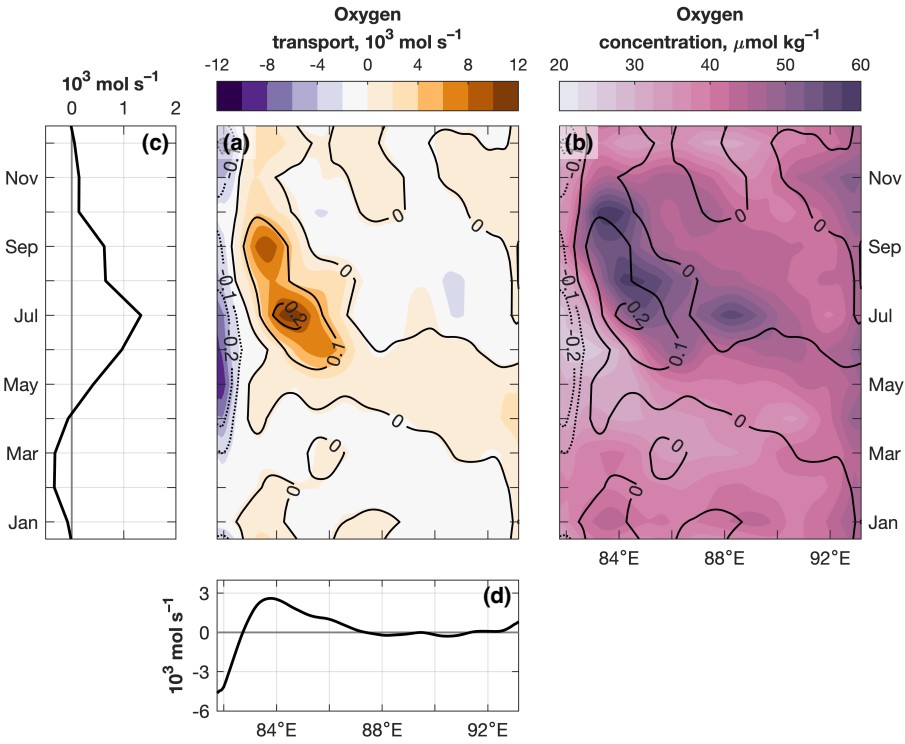

**Figure 4.** Annual cycle of **(a)** meridional oxygen transport ($T_{O_2}$; Equation 1; $10^3$ mol s$^{-3}$) and **(b)** oxygen concentration ($\mu$mol kg$^{-1}$) at 8°N between the 24 and 26 kg m$^{-3}$ isopycnals. Contours indicate meridional velocity (m s$^{-1}$); positive contours are solid, negative contours are dotted. In panels **(c)** and **(d)**, the oxygen transport plotted in panel (a) is averaged over longitude and time respectively.

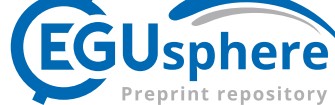

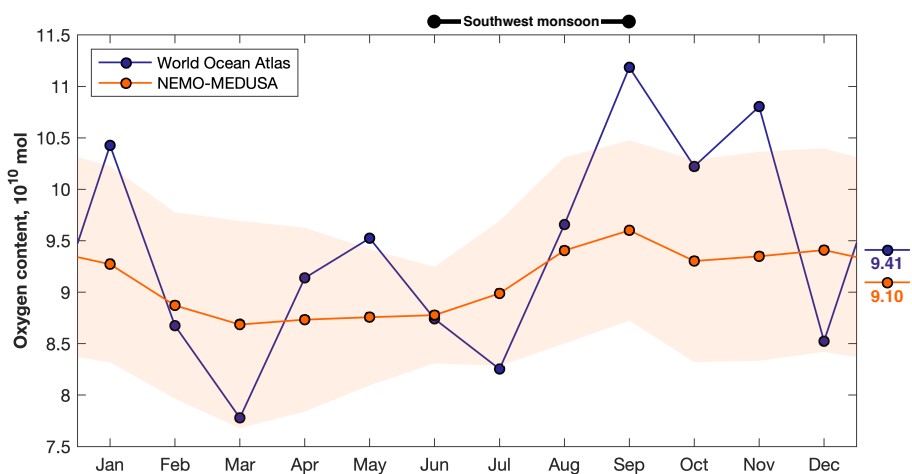

**Figure 5.** Monthly oxygen content ($10^{10}$ mol), from the World Ocean Atlas (in blue) and NEMO-MEDUSA (in orange), of the Bay of Bengal oxygen minimum zone, north of 8°N, between the 24 and 26 kg m$^{-3}$ isopycnals. The shaded region indicates ± one standard deviation (NEMO-MEDUSA only); standard deviations are are not plotted for World Ocean Atlas observations, which are available only as climatological monthly means. The annual mean oxygen content ($10^{10}$ mol) from the World Ocean Atlas (blue) and NEMO-MEDUSA (orange) is shown to the right of the plot; the period of the southwest monsoon is indicated by the bar above the plot.

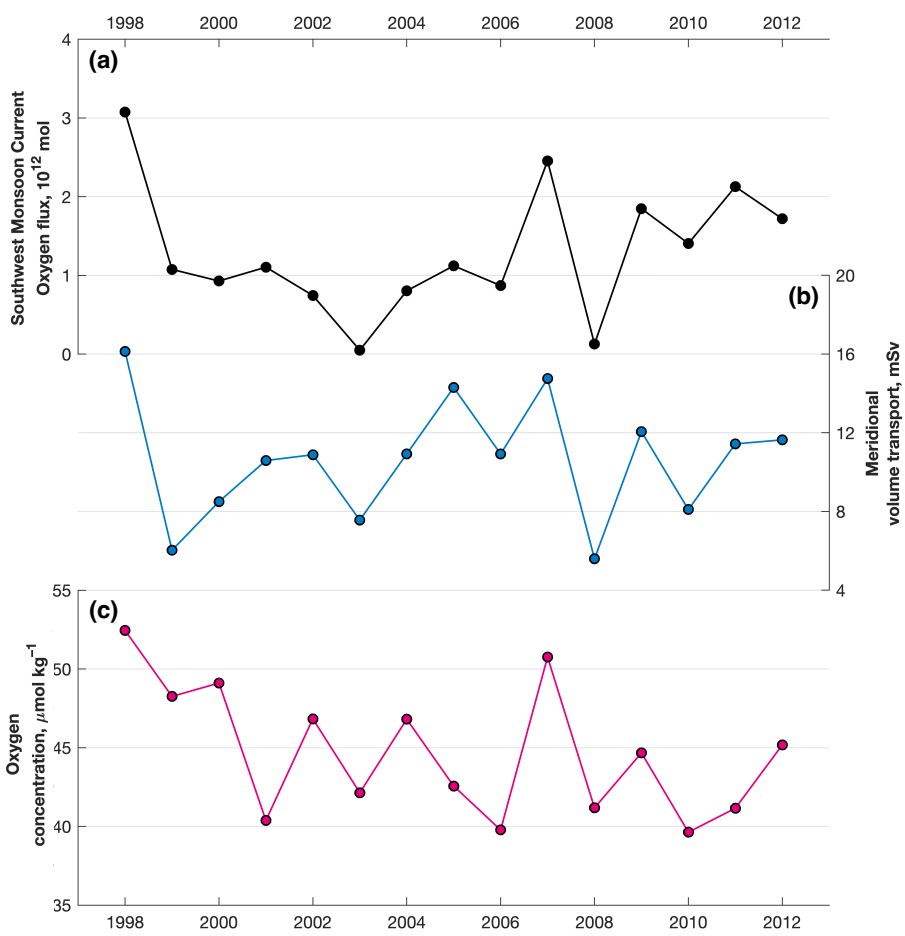

**Figure 6. (a)** Flux of oxygen ($10^{12}$ mol) integrated over the four months of the southwest monsoon (i.e. June to September), calculated from NEMO-MEDSUA (solid black line). **(b)** Annual-mean volume transport (mSv) during the southwest monsoon. **(e)** Annual-mean oxygen concentration during the southwest monsoon ($\mu$mol kg$^{-1}$). All quantities are calculated at $8°$N, between the 24 and 26 kg m$^{-3}$ isopycnals

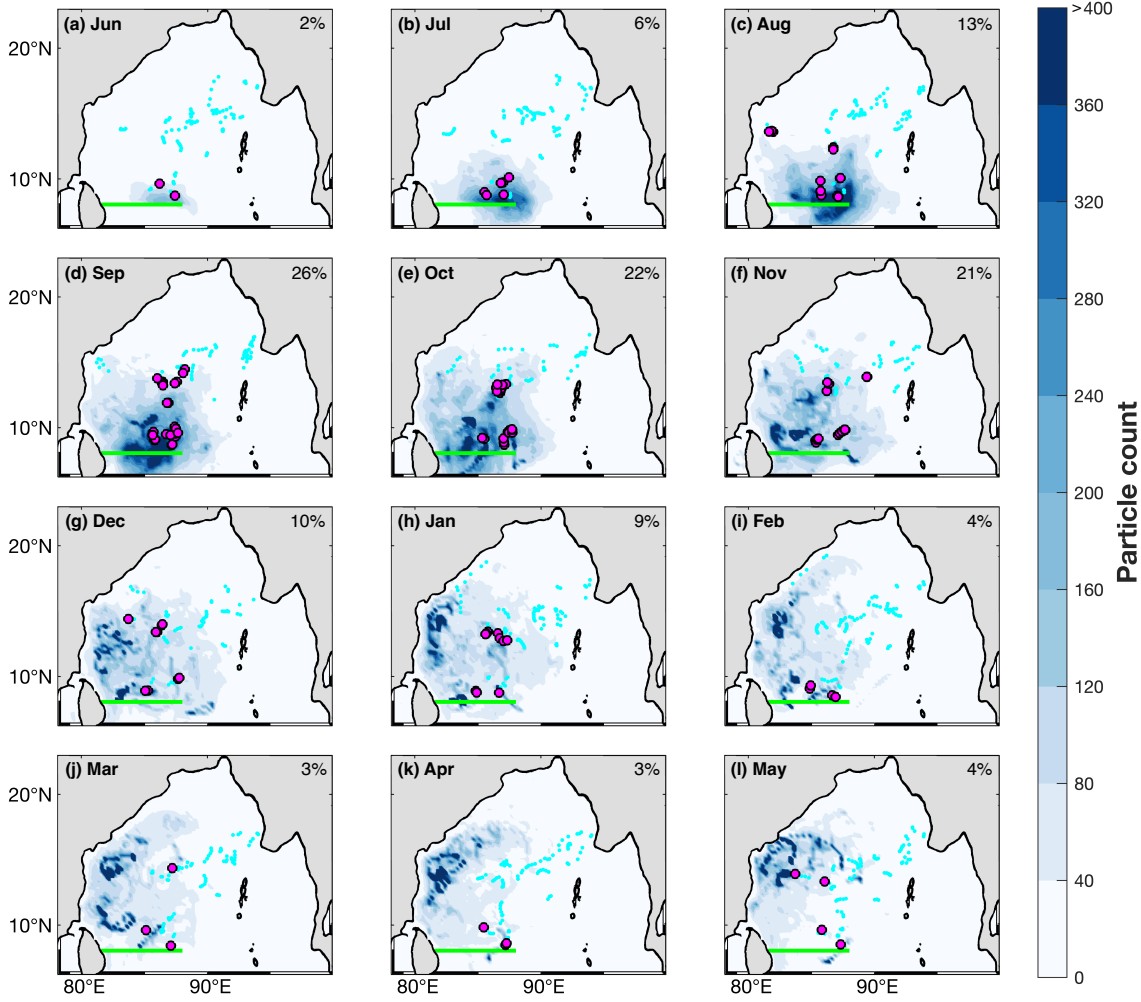

**Figure 7.** Monthly mean particle count, in 0.25° bins, from the forwards-trajectory experiments. Particles were released at 8°N at 0.01° intervals between the Sri Lankan coast and 88°E (i.e. along the green line) where and when salinity is greater than 35.2 PSU; releases are daily between 1 June and 30 September (inclusive; 1994 to 2019) and particles are tracked forward for a year; see Section 2.1 for details. (Note that the GLORYS12 re-analysis outputs practical salinity in PSU.) Small, light blue dots indicate locations of all bio-Argo profiles in each month; large pink dots indicate profiles with a salinity-oxygen spike indicative of Southwest Monsoon Current water at the 24 kg m$^{-3}$ isopycnal; see Section 2.1 for details. The percentage of Argo profiles with a high-salinity, high-oxygen spike is shown in the top right corner of each panel.

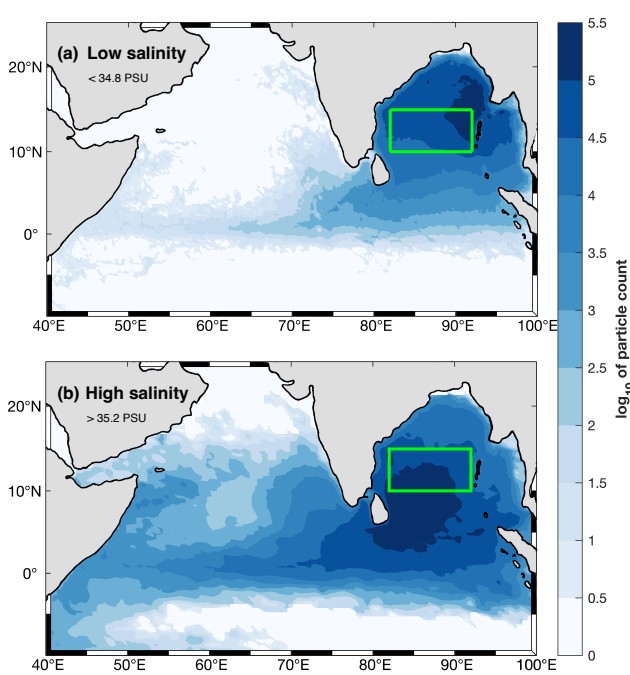

**Figure 8.** Logarithm (base 10) of particle count, in 0.25° bins, from the backwards-trasjectory experiments. Particles were released within the green-outlined box where **(a)** salinity was less than 35 PSU, and **(b)** salinity was greater than 35.2 PSU. (Note that the GLORYS12 re-analysis outputs practical salinity in PSU.)