# Peer review of "Ventilation of the Bay of Bengal oxygen minimum zone by the Southwest Monsoon Current"

_EGUsphere, 2024_

## Author Response (AR1)

**Response to reviews**

Throughout this document, the original reviews are reproduced in black. Our responses are presented in blue and, where appropriate, quotations from the revised paper are included in indented *italic blue* text. Please note that line numbers refer to the tracked changes version of the manuscript.

**Reviewer One**

The authors present glider observations, over 17 days, of an oxygen-rich core to the Arabian Sea High-Salinity Water (ASHSW) as it flows into the Bay of Bengal during the 2016 southwest monsoon. Using a global climatology, Argo floats and a coupled physical-biogeochemical model, they estimate the contribution of these waters to the Bay of Bengal oxygen minimum zone.

The fact the ASHSW appears to ventilate and alleviate oxygen reduction appears to be contrary to previous studies and represents an important new observation.

These observations are in agreement with the analysis of Argo float data in this preprint: https://arxiv.org/abs/2406.10571

We thank the reviewer highlighting for this paper. As it is now published, we have been pleased to add references to it in the manuscript where relevant.

The authors highlight large discrepancies (> 40 µmol kg$^{-1}$) between the world ocean atlas and the model over large parts of the bay.

The paper is well written, clear and concise. The figures are high quality and legible. The limitations of the study are related to the quality of the publicly available climatologies for the region. It is a shame uncertainties are not available.

We thank the reviewer for their positive feedback.

**Comments**

While in virtual mooring mode, how close to their nominal position do the gliders remain during the study? I.e. how much of the variance see in figure 3 could be horizontal variability?

When in virtual mooring mode, the gliders did not travel particularly far from the central point / target position, moving on average between 2 and 3 km over the course of a single dive and remaining on average 7 km from the waypoint. These distances are typical for Seagliders diving to 700 m, as was

largely the case for this deployment. Of course, these distances are negligible compared with those between the gliders (i.e. around 110 km).

It is always the case with glider observations that spatial and temporal variability cannot ever be completely separated – so, for instance, the sharp transition from high-salinity water to low-salinity water (and back again) observed by SG534 could contain a degree of spatial as well as temporal variability. We note, however, that this still supports our interpretation of this feature as a sharp front: either a front that passes over a largely stationary glider or a stationary front that the glider itself crosses. While it is reasonable to claim that the hydrographic differences between different gliders are spatial, we are careful not to ascribe hydrographic differences observed by the same glider to either spatial or temporal variability.

> **Line 73.** *The gliders remained an average of 7 km from the exact location of the virtual mooring, and travelled an average of 2 to 3 km during a single dive; this is typical for gliders diving to 700 m, as was largely the case during this deployment. We consider each glider's observations as a time series, although a very small degree of spatial variability may be present in an individual glider's record.*

65 – How were the glider sensors calibrated / validated?

Originally, the oxygen observations were calibrated using manufacturer coefficients, as well as using a sodium sulphite solution to calibrate the zero-concentration end point. Given that Nayak et al. (2025) has now been published – the pre-print highlighted by the reviewer above – we may now use their Winkler-calibrated CTD observations, which were previously unavailable. In order to provide greatest consistency between papers and instruments, we have performed a new calibration of glider oxygen. Glider oxygen concentrations were linearly regressed against CTD oxygen concentrations in temperature-salinity space to avoid variability induced by internal waves or oxygen gradients along isopycnals. The new calibrations provide only a minor change to oxygen calibrations and do not qualitatively affect our conclusions, which were largely based on the gradients in oxygen between the gliders rather than absolute values.

> **Line 77.** *Oxygen sensors were calibrated using a sodium sulphite solution to find the zero-concentration end-point; subsequently, we calibrated glider-observed oxygen concentration against the Winkler-calibrated ship observations of Nayak et al. (2025). We linearly regressed glider oxygen concentrations against ship oxygen concentrations in temperature-salinity space to avoid variability induced by internal waves or oxygen gradients along isopycnals. This second step results in only a minor change in oxygen concentrations.*

130 – Is it possible to estimate the impact of ignoring vertical processes?

Unfortunately, we think that it would be very difficult to quantify the impact of ignoring vertical processes – e.g. diapycnal mixing – on the results of the particle tracking experiments. The assumption behind the particle tracking experiment is that the 24 kg m$^{-3}$ isopycnal remains the core of the ASHSW layer as the Southwest Monsoon Current flows northward into the Bay of Bengal. Real-world tracer-release experiments find that the density at which a tracer is released into the ocean remains the central density of the tracer distribution even after a year (Ledwell et al., 2011). In our case, given the presence of high-salinity, high-oxygen spikes found on this isopycnal in the Argo observations, this assumption does not appear to be unreasonable. What is more, the 24 kg m$^{-3}$ isopycnal represents the core of ASHSW at its formation in the Arabian Sea (e.g. Prasana Kumar & Prasad, 1999): this is still the case in the Bay of

Bengal. We further assume that diapycnal mixing would primarily act to reduce the magnitude of the signal without significantly altering the trajectories of the particles; in other words, we assume that the isopycnal pathways would not be significantly altered by a small amount of diapycnal mixing.

To test this assumption, we conducted forwards particle tracking experiments, identical to those originally conducted on the 1024 kg m$^{-3}$ isopycnal, on the 1024.5 and 1025 kg m$^{-3}$ isopycnals. The results of these experiments are presented at the end of this document, in Figures A and B respectively. While there are small differences between the results of the three experiments, the distribution of particles tracked on the three isopycnals is broadly consistent. Hence, although in the real ocean diapycnal mixing would be expected to spread some portion of ASHSW away from the 1024 isopycnal, this process appears unlikely to result in a markedly different distribution of ASHSW. We have added this point to the paper.

> *Line 153. This method determines advective transport along an isopycnal and does not account for isopycnal or diapycnal diffusivity; however, we note that the results presented below are qualitatively very similar to those from versions of the same experiments conducted on the 24.5 and 25 kg m–3 isopycnals.*

211 – Is the SMC the only source of water with salinity > 35.2?

We thank the reviewer for this pertinent question. In considering our response, we have chosen to lower the threshold from 35.2 to 35.1 PSU, to better represent the SMC in salinity at 8°N. We note that this results in more particles being released, but does not make a qualitative difference to our results, nor does it change any of our conclusions. In particular, the match between the particle tracking results and the Argo profiles with high-salinity, high-oxygen spikes is unchanged. (For consistency, we have also reduced the threshold to 35.1 PSU in the backwards tracking experiment; again, this makes no qualitative difference to our results or conclusions.)

At the bottom of this response document, we attach a Hovmöller plot of the annual cycle of salinity at 8°N from the GLORYS12 ocean re-analysis (Figure C). Salinities greater than 35.1 PSU are found at 8°N from June between 85 and 86°E before moving generally westward as the southwest monsoon progresses. This resembles the velocity (and oxygen) signal of the SMC – compare Figure 4 of the revised paper. Furthermore, salinities greater than 35.1 PSU are not found elsewhere at 8°N – i.e. outside the SMC. (Note that we release particles from June to September, so the limited incursion of high-salinity water in late January will not affect the particle tracking experiments.)

**Technical**

114 – awkward sentence, in addition -> similarly

This sentence has been amended.

260 – "in the ASHSW core oxygen concentration"

This sentence has been amended.

> *Line 298. Consequently, it appears that processes which drive variability in the oxygen concentration of ASHSW have only a modulating effect on the SMC's oxygen flux, and that*

*the predictability of the SMC's oxygen flux derives primarily from understanding and predicting the SMC's physics.*

278 – "than those considered here"

This sentence has been amended.

> **Line 315.** *Constructing an oxygen budget for the Bay of Bengal is beyond the scope of the present contribution – clearly this will depend on many more processes, both physical and biogeochemical, than on those considered here.*

**References**

Prasana Kumar S and Prasad TG, 1999. Formation and spreading of Arabian Sea high-salinity water mass. *Journal of Geophysical Research*, 104, 1455 – 1464

Ledwell JR, St Laurent LC, Girton JB, Toole JM, 2011. Diapycnal mixing in the Antarctic Circumpolar Current. *Journal of Physical Oceanography*, 41, 241 – 246

[Figure]

**FIGURE A** Monthly mean particle count, in 0.25° bins, from the forwards trajectory experiments. Particles were released on the **1024.5 kg m⁻³** isopycnal at 8°N at 0.01° intervals between the Sri Lankan coast and 88°E (i.e. along the green line) where and when salinity is greater than 35.1 PSU; releases are daily between 1 June and 30 September (inclusive; 1994 to 2019) and particles are tracked forward for a year.

[Figure]

**FIGURE B** Monthly mean particle count, in 0.25° bins, from the forwards trajectory experiments. Particles were released on the **1025 kg m⁻³** isopycnal at 8°N at 0.01° intervals between the Sri Lankan coast and 88°E (i.e. along the green line) where and when salinity is greater than 35.1 PSU; releases are daily between 1 June and 30 September (inclusive; 1994 to 2019) and particles are tracked forward for a year.

[Figure]

**FIGURE C** Hovmöller plot of the annual cycle of salinity (PSU), at daily resolution, at 8°N in the Bay of Bengal from the GLORYS12 re-analysis product. The 35.1 PSU isohaline is plotted in black.

**Response to reviews**
* * *
Throughout this document, the original reviews are reproduced in black. Our responses are presented in blue and, where appropriate, quotations from the revised paper are included in indented *italic blue* text. Please note that line numbers refer to the tracked changes version of the manuscript.

**Reviewer Two**

The Oxygen Minimum Zone in the Bay of Bengal is at the threshold to de-nitrification, which would lead to increased production and release of climate-relevant trace gases. To understand the processes that have an impact on development and maintenance of the OMZ, a better knowledge of the ventilation pathways into the OMZ is crucial.

The focus of this study is on the important role of the Southwest Monsoon Current, which transports oxygen-rich Arabian Sea High Salinity Water into the Bay of Bengal. The supply of oxygen via the SMC could be a key point in the ventilation of the OMZ in the Bay of Bengal. The study is based on multi-platform observations data consisting of four gliders, WOA climatology, satellite data, and Bio-Argo floats as well as a biogeochemical model.

The study is well written and structured and the topic highly relevant.

We thank the reviewer for their positive feedback.

I recommend publication of this manuscript after minor revision. I will leave my comments below in the order in which they appear in the text:

**Comments**

Line 64–66: Please add when (month, year) the gliders were deployed?

This has been added.

> **Line 69.** *The gliders were deployed in early July 2016, east to west along 8°N, 70 at 86, 87, 88 and 89°E. Deployed at 86°E, SG579 transited to 85.3°E after seven days (38 dives), where it stayed for the remaining 12 days of the deployment (78 dives). Otherwise, the gliders were operated as "virtual moorings": i.e. they remained on-station for the entire deployment (between 10 and 17 days).*

Line 75: SEALEVEL_GLO_PHY_L4_NRT_008_046 – this dataset provides near-real time data for the period of 2022 to 2025, whereby the data used in Fig. 1 show July 2016. You probably might have used dataset SEALEVEL_GLO_PHY_L4_MY_008_047 for the period 1993 to 2023?

We apologise for this mistake. The correct dataset is now referenced.

> *Line 87. Satellite altimetry observations of surface velocity, provided by the Copernicus Marine Data Service (Product ID: SEALEVEL_GLO_PHY_L4_MY_008_047).*

Chapter 2.2.1: Perhaps it's worth pointing out that the SMC core from observations (Fig. 1) is significantly further east compared to the model data (Fig. 2b). This would also explain, why the oxygen maximum from glider observations (Fig. 3) is also further east compared to the model data (Fig. 4b).

This is a very good point. We have added a comment on this to the manuscript.

> *Line 126. The SMC is present as a northward flow in the southwestern Bay of Bengal, albeit some two to three degrees further west than its position during the period of the observations presented here (Figs. 1 and 2b); this same westward offset in the model is also apparent when compared with the glider observations (Fig 3).*

Line 111: It should read Fig. 2a (instead of Fig. 1a).

Corrected.

Line 143: Please indicate Fig. 3a

Corrected.

Line 146: "The gradual and steady decrease…" it should read "increase"

Corrected.

Fig. 3: These plots show very nicely the SMC core of younger, high-salinity, and oxygen-rich ASHSW in comparison to the ambient water of the OMZ. I think the results could be better presented by enlarging the y-axis as the contour lines are very close to each other.

We thank the reviewer for this excellent suggestion – the figure has been re-plotted with larger y-axes.

Fig. 6: I am probably missing something here (apologizes), but it is not entirely clear to me why the oxygen flux can be zero in 2003 (Fig. 6a), while the meridional volume transport and the oxygen concentration are both positive (Fig. 6b, c).

We thank the reviewer for pointing this out. For Figure 6, in the original draft of the manuscript, we calculated the oxygen flux and the spatio-temporal integral of the oxygen flux over the four months of each year's southwest monsoon: i.e. integrating between the 24 and 26 kg m$^{-3}$ isopycnals, across the width of the Bay of Bengal, and from June to September. The volume transport was the integrated volume transport between the 24 and 26 kg m$^{-3}$ isopycnals and across the width of the Bay of Bengal, *averaged* from June to September. Both quantities included both in- and outflow – and in the case of volume transport this resulted in a very small value indeed (presented in units of mSv, i.e. 10$^{-3}$ Sv). Oxygen concentration was simply the spatio-temporal mean between the 24 and 26 kg m$^{-3}$ isopycnals, across the width of the Bay of Bengal, and between June and September.

We have carefully checked our original calculations and we believe them to be correct. We wonder whether the discrepancy pointed out by the reviewer arises because, numerically, the product of the mean of A and the mean of B is not necessarily the same as the mean of AB; for example, if the small positive net inflow was comprised of a large inflow of low-oxygen water and a slightly smaller outflow of high-oxygen water, then the integral of the oxygen transport could actually be negative despite both averages being positive. However, this is clearly neither intuitive nor physically satisfying. Furthermore, the net transport is of course the balance between inflow and outflow; this is always much smaller than the SMC transport and, indeed, is always close to zero. On reflection, this is not the best way of characterising the strength of the inflow.

We have revised our calculations and have re-plotted Figure 6. The oxygen and volume transports are now calculated as *spatial integrals* (i.e. between the 24 and 26 kg m$^{-3}$ isopycnals and across the width of the Bay of Bengal) and then as *temporal means* (i.e. from June to September). Critically, we now exclude model grid points where velocity is negative, hence we include only inflow. Oxygen concentration is a spatio-temporal mean, as before, but now also with the new velocity constraint. These revised time series avoid the apparent conflict raised by the reviewer, as well as better isolating just the inflow to the Bay of Bengal that is the focus of our study. (To demonstrate that there is indeed a net inflow of oxygen between the 24 and 26 kg m$^{-3}$ isopycnals, we also present oxygen transport calculated without the velocity constraint.)

> ***Line 114.*** *To investigate interannual variability, we need an average value of* $T_{O2}$ *for each year's southwest monsoon. We integrate* $T_{O2}$ *over the width of the Bay of Bengal: firstly including all grid points across the Bay; and secondly, to better approximate the oxygen transport of the northward-flowing SMC, including only those grid points with a northward meridional velocity. We then average the integrated oxygen transport over the four months of the southwest monsoon to derive a mean value for each year. Similarly, we calculate the average northward volume transport across 8°N, integrating between the 24 and 26 kg m$^{-3}$ isopycnals and across the width of the Bay, then averaging over the four months of the southwest monsoon. Finally, we calculate the mean oxygen concentration across the Bay, including only model grid cells with a northward velocity, and average over the four months of the southwest monsoon.*

Figure caption of Fig. 7, line 3: "section 2.1" this should read 2.3.

Corrected.

Line 258: It should read Fig. 4 instead of Fig. 1.

Corrected.